# A Novel Confidence Guided Training Method for Conditional GANs with Auxiliary Classifier

## ABSTRACT

Conditional Generative Adversarial Network (cGAN) is an important type of GAN which is often equipped with an auxiliary classifier. However, existing cGANs usually have the issue of mode collapse which can incur unstable performance in practice. In this paper, we propose a novel stable training method for cGANs with well preserving the generation fidelity and diversity. Our key ideas are designing efficient adversarial training strategies for the auxiliary classifier and mitigating the overconfidence issue caused by the cross-entropy loss. We propose a classifier-based cGAN called Confidence Guided Generative Adversarial Networks (CG-GAN) by introducing the adversarial training to a $K$-way classifier. In particular, we show in theory that the obtained $K$-way classifier can encourage the generator to learn the real joint distribution. To further enhance the performance and stability, we propose to establish a high-entropy prior label distribution for the generated data and incorporate a reverse KL divergence term into the minimax loss of CG-GAN. Through a comprehensive set of experiments on the popular benchmark datasets, including the large-scale dataset ImageNet, we demonstrate the advantages of our proposed method over several state-of-the-art cGANs.

## CCS CONCEPTS

• **Computing methodologies** → **Computer vision**.

## KEYWORDS

Image generation, Conditional generative adversarial network

## 1 INTRODUCTION

Generative Adversarial Network (GAN) [7] is a popular generative model for high-fidelity image generation, which has been extensively studied in recent years [1, 18, 27, 33]. Though other generative models, such as the diffusion models [4, 10], have recently also attracted a lot of attentions due to their effectiveness in generating high-quality images, GANs still enjoy several significant advantages in practical applications, such as their lower computational complexities for training and inference [14, 15]. The key idea of GAN is to simultaneously train a generator and a discriminator by using the adversarial game: the generator takes random noise to generate the fake data so as to fool the discriminator, meanwhile, the discriminator tries to distinguish between the real and fake data.

During this adversarial training process, the generator becomes stronger and can generate new data (e.g., images) with high quality. The original GANs do not utilize the label information (e.g., the labels of the train data). Mirza and Osindero [26] proposed the **conditional GANs (cGANs)** that utilize the label information to generate some specific class of data. For example, one can condition the cGANs with the labels for animals to generate the images of dogs and cats separately. cGANs have been used for various applications, such as text-to-image generation [15, 31], image style transformation [39], and speech enhancement [25].

In general, most existing cGANs can be classified into two main categories, the classifier-based cGANs [2, 11, 30] and the projection-based cGANs (which incorporate the conditional information into the discriminator) [8, 28]. For a classifier-based cGAN, it often takes advantage of a classifier to utilize the class information. For example, the classifier can penalize the mismatched data-label pairs during the training process [11, 13]. As a representative classifier-based cGAN, the "Auxiliary classifier GAN (AC-GAN)" proposed by Odena et al. [30] contains an auxiliary classifier to learn a conditional label distribution for guiding the generator to generate class-specific images. Although these proposed cGANs can achieve promising generation quality, recent researches have shown that they often suffer from two problems in practice: **(1)** the performance of the generator drops at the early training stage (i.e., early-training collapse), particularly when working with datasets that have a large number of classes[11, 13, 28, 37]; **(2)** the generator tends to generate data with low diversity [11, 28].

To improve the performance of cGANs, a number of elegant methods have been proposed, which mainly focus on modifying the network structure or the loss function of the classifier. For example, to remedy the early-training collapse issue, ReACGAN [13] normalizes both the input feature vectors and the weight vectors in the classifier; Hou et al. [11] proposed ADC-GAN that is based on an auxiliary discriminative classifier for achieving better training stability and generation diversity (we provide a detailed introduction on more existing approaches in Section 5). Despite of the improvements achieved by these methods, their practical performances are still not quite satisfying in some scenarios. For example, in Figures 1a and 1b, we illustrate the Inception Score (IS) [32] and Fréchet Inception Distance (FID) [9] curves of AC-GAN and several improved models on Tiny-ImageNet [23], where IS and FID are commonly used metrics for evaluating the performance in terms of generation fidelity and diversity. We can see that the performances of some cGAN methods decrease after a certain number of iterations. In Figure 1c, we show the classification accuracies on generated images of several cGAN methods on the large-scale dataset Imagenet [3]; some of their conditional generation performances are relatively low.

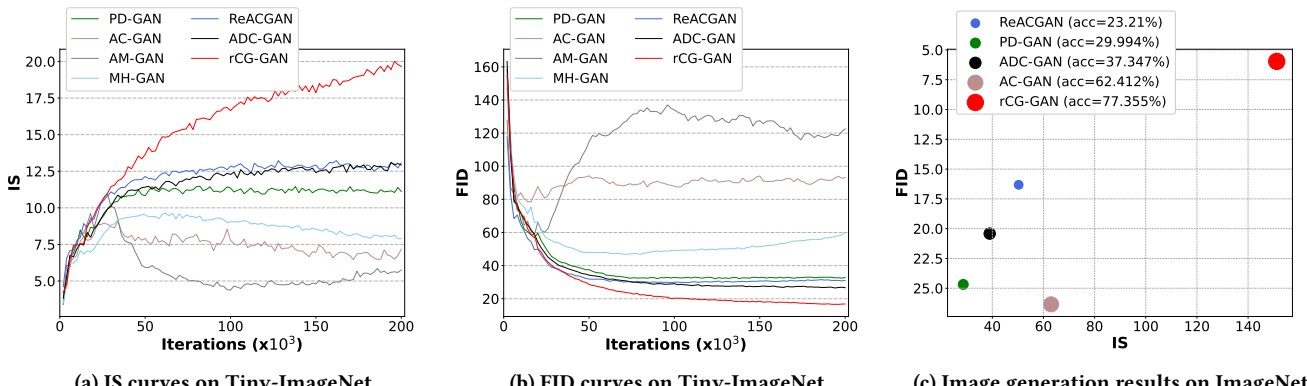

(a) IS curves on Tiny-ImageNet.

(b) FID curves on Tiny-ImageNet.

(c) Image generation results on ImageNet.

Figure 1: (a) and (b) are the IS (higher is better) and FID (lower is better) curves on Tiny-ImageNet [23]. The figures contains the curves of PD-GAN [28], AC-GAN[30], AM-GAN [38], Multi-hinge (MH) GAN [20], ReACGAN [13], and ADC-GAN [11]; the "rCG-GAN" is our proposed cGAN method in Section 4. (c) Conditional image generation results on ImageNet [3]. The "acc" (higher is better) means the ImageNet classification top-1 accuracy on the generated images, which reveals the conditional generation performance. The numerical results of AC-GAN, PD-GAN and ReACGAN are reported from [13].

**Our contributions.** From the above discussion and the experimental results shown in Figure 1, we can see that the major challenge for designing a promising cGAN is to **achieve stable and high-quality generation as well as high conditional generation performance**. We design an efficient adversarial training strategy for the auxiliary classifier and propose a novel conditional GAN method based on the aforementioned AC-GAN, which is called "Confidence Guided Generative Adversarial Networks (CG-GAN)".

We demonstrate theoretically that the $K$-way classifier of CG-GAN can encourage the generator to approach the real joint distribution. By analyzing the gradient of the classification loss in CG-GAN, we elucidate the mechanism through which our CG-GAN improves training stability. Furthermore, we investigate the potential challenges that CG-GAN may encounter in practical applications. To mitigate these issues, we propose the establishment high-entropy prior label distribution for the generated data. Subsequently, we incorporate the corresponding inverse Kullback-Leibler (KL) divergence term into the minimax loss of CG-GAN, leading to enhanced stability and performance. We refer to the CG-GANs with reverse KL divergence terms as "rCG-GAN". The potential limitations of the two most closely related works are analyzed, (i.e., ReACGAN [13] and ADC-GAN [11]), to highlight the advantage of rCG-GAN.

To validate the effectiveness of our method, a set of experiments on several popular banchmark datasets are conducted, including CIFAR10, CIFAR100 [21], Tiny-ImageNet [23], Baby/Papa/Grandpa-ImageNet [14] and ImageNet [3]. Compared with the recently proposed classifier-based and projection-based GANs, our rCG-GAN can achieve better performance in terms of the IS and FID scores over those benchmark datasets. Notably, for the large-scale dataset ImageNet, rCG-GAN can achieve significant improvement over the previous state-of-the-art methods: rCG-GAN yields the IS that is at least two times of their IS scores, and the FID that is only half of theirs. Moreover, rCG-GAN has better top-1 and top-5 classification accuracies on generated images, which indicate the improvement on conditional generation performance.

**The rest of this paper is organized as follows**. In Section 2, we overview the definitions for GAN and the related cGANs with auxiliary classifier. We conclude that adversarial training is often missing applied to the auxiliary classifier in these cGAN models. Then, we explore the relationship between early-training collapse and over-confidence. In Section 3, we show our method for training classifier-based cGANs. In Section 4, we illustrate our experimental results and the comparisons with several state-of-the-art cGANs. Finally, we discuss the related work and conclude in Section 5 and Section 6, respectively.

## 2 PRELIMINARIES

**Generative Adversarial Networks.** Let $X$ be the data space. The original GAN [7] consists of two neural networks: the generator $G$ that maps a given random noise $z$ to a generated data point $x \in X$, and the discriminator $D$ that distinguishes between real data and generated data by mapping each data point $x \in X$ to a value in $[0, 1]$. Denote by $P_X$ and $Q_X$ the real data distribution and the generated data distribution, respectively. The adversarial training is to optimize the following losses:

$$\min_D L_D = -\mathbb{E}_{x \sim P_X}\left[\log D(x)\right] - \mathbb{E}_{x \sim Q_X}\left[\log\left(1 - D(x)\right)\right]; \quad (1)$$

$$\min_G L_G = \mathbb{E}_{x \sim Q_X}\left[\log\left(1 - D(x)\right)\right]. \quad (2)$$

**AC-GAN.** The Auxiliary classifier GAN (AC-GAN) [30] is one of the most representative Classifier-based cGANs, which uses an auxiliary classifier to improve the performance of the ordinary GAN. The objective of AC-GAN also consists of two parts as the GAN losses Equation (1) and Equation (2), where the difference is that they both contain the penalty items for the classification loss.

Given the training dataset with $K$ classes ($K \in \mathbb{Z}^+$), a $K$-way classifier "$l$" maps each input $x \in X$ to the label space $\mathbb{R}^K$. Let $Y = \{1, 2, \cdots, K\}$. For each $y \in Y$, let $l_y(x)$ denote the $y$-th element of $l(x)$ corresponding to the label $y$. A widely used loss for classifier

is the **softmax cross-entropy** loss with one-hot encoding:

$$\sigma_{\mathsf{sce}}(x, y) = -\log \Pr(y|x) = -\log \frac{\exp\left(l_y(x)\right)}{\sum_{k=1}^{K} \exp\left(l_k(x)\right)}, \quad (3)$$

where $\Pr(y|x)$ is the conditional probability of the ground-truth label $y$. For ease of presentation, we in particular call $\Pr(y|x)$ the "**confidence**" of $y$; namely, the confidence indicates the probability that $x$ belongs to class $y$. In Section 3, we will introduce our cGAN model based on $\Pr(y|x)$, and that is why we name it as "confidence guided cGAN". Let $P_{XY}$ (resp., $Q_{XY}$) denote the joint distribution of the real (resp., generated) data and labels in $X \times Y$. The objective functions of AC-GAN are defined as follows:

$$\min_{D} L_D + \lambda \cdot \mathbb{E}_{x, y \sim P_{XY}} \left[\sigma_{\mathsf{sce}}(x, y)\right]; \quad (4)$$

$$\min_{G} L_G + \lambda \cdot \mathbb{E}_{x, y \sim Q_{XY}} \left[\sigma_{\mathsf{sce}}(x, y)\right], \quad (5)$$

where $L_D$ and $L_G$ are defined in Equation (1) and Equation (2), and $\lambda > 0$ is the given coefficient.

**Adversarial training missing on the auxiliary classifier.** MH-GAN [20] improves upon AC-GAN by substituting the cross-entropy loss with a multi-class extension of the widely used hinge loss. ReACGAN [13] suggests normalizing feature embeddings onto a unit hypersphere to address the early-training collapse problem, and expanded the cross-entropy loss of the classifier to the data-to-data cross-entropy loss. We conclude that most classifier-based cGANs, such as AC-GAN, MH-GAN and ReACGAN, lack efficient adversarial training on the auxiliary classifier. This deficiency makes it challenging for generators to learn the real joint distribution of the training data, thereby diminishing the diversity of the generated samples.

**Early-training collapse and over-confidence.** As mentioned before, conditional GANs with auxiliary classifier are prone to early-training collapse. An important observation of Kang et al. [13] is that the unboundedness of input feature norm can cause undesirable gradient explosion problem for the classifier of AC-GAN, which usually leads to early-training collapse. Also, as discussed in several papers [16, 35], large feature norm is a main reason for over-confidence (i.e., peaky prediction distribution), because the confidence is usually proportional to its feature norm and it turns to encourage the classifier to always output increasingly large feature norm so as to encourage high confidence.

Based on these insights, in the Section 3, we focus on designing novel classification loss functions which incorporate effective adversarial training strategies and addressing the issues of early-training collapse and over-confidence.

## 3 OUR PROPOSED TRAINING METHOD

In this section, we propose a novel stable adversarial training method for the classifier-based cGANs. First, we introduce our basic model "**CG-GAN**", and explain that why CG-GAN can encourage the generator to learn the real joint distribution optimally in Section 3.1. Then, we study the gradient of the classification loss of CG-GAN in Section 3.2, which is the key to improve the training stability. In Section 3.3, we discuss the challenges for training the basic CG-GAN and propose an improved version "rCG-GAN", which can be implemented more efficiently in practice. Finally, we discuss distinctions between rCG-GAN and closely related works,

analyzing potential limitations of ReACGAN and ADC-GAN [11] to highlight advantages of rCG-GAN.

### 3.1 CG-GAN

**Our high-level idea.** To introduce the adversarial training to a classifier, a natural idea is to encourage the discriminator to return higher confidence for the output from real data and lower confidence for the output from generated data. This intuition is somewhat similar to the adversarial strategy for training a standard GAN [7]. However, we focus on the classifier and need to develop some significant new ideas for the loss function with the analysis from the stability perspective. The key of implementing our idea is to design an appropriate classification loss for achieving an effective balance between the adversarial training and the confidence.

In the stage of optimizing the discriminator, we consider minimizing the softmax cross-entropy loss (i.e, $\sigma_{\mathsf{sce}}(x, y)$ in Equation (3)) on real data and maximizing $\sigma_{\mathsf{sce}}(x, y)$ on generated data. Specifically, we intend to minimize the following classification loss in the training procedure for the discriminator:

$$\mathbb{E}_{x, y \sim P_{XY}} \left[\sigma_{\mathsf{sce}}(x, y)\right] - \mathbb{E}_{x, y \sim Q_{XY}} \left[\sigma_{\mathsf{sce}}(x, y)\right]. \quad (6)$$

It is worth noting that directly optimizing the loss Equation (6) may result in a technical issue in practice: the value of $\sigma_{\mathsf{sce}}(x, y)$ on generated data can be quite large, making the training process challenging to converge and undermining the performance of the classifier. To avoid this issue, we incorporate a hinge loss and introduce an upper bound "$m > 0$" to our model. Since "$\sigma_{\mathsf{sce}}(x, y)$" depends on the confidence $\Pr(y|x)$ (see Equation (3)) and the introduced parameter $m$ can restrict the confidence, we name this model as "Confidence Guided Generative Adversarial Network **(CG-GAN)**". Formally, the objective functions for the classifier of CG-GAN in the discriminator and generator are defined as follows:

$$C_d^{cg} = \mathbb{E}_{x, y \sim P_{XY}} \left[\sigma_{\mathsf{sce}}(x, y)\right] + \mathbb{E}_{x, y \sim Q_{XY}} \left[\left[m - \sigma_{\mathsf{sce}}(x, y)\right]_+\right]; \quad (7)$$

$$C_g^{cg} = \mathbb{E}_{x, y \sim Q_{XY}} \left[\sigma_{\mathsf{sce}}(x, y)\right], \quad (8)$$

where the notation $[a]_+ = a$ if $a \geq 0$, otherwise, $[a]_+ = 0$. The second term of Equation (7) serves as a crucial aspect during optimization, indicating that when the value of $\sigma_{\mathsf{sce}}(x, y)$ on generated data exceeds the value of $m$, it will not further increase. This thresholding mechanism helps to mitigate the issue of degradation of classifier performance, a potential challenge that can be encountered in adversarial training settings. Coupled with the GAN losses, our CG-GAN has the following objectives:

$$\min_{D} L_D + \lambda \cdot C_d^{cg}; \quad (9)$$

$$\min_{G} L_G + \lambda \cdot C_g^{cg}. \quad (10)$$

We propose the following Proposition 3.1, which states the conditions under which the training objective of the classifier of CG-GAN achieves the global optimum.

PROPOSITION 3.1. *The global optimum of the training objective for the classifier of CG-GAN can be achieved if and only if $Q_{XY} = P_{XY}$.*

Proposition 3.1 reveals that the classifier encourages the generator to learn the real joint distribution. Due to the space limit, we place the proof to our supplement.

## 3.2 Analysis on The Gradient of The Classifier

As discussed in Section 2, a major reason for early-training collapse and over-confidence is from the unboundedness of feature norm and its induced gradient explosion problem. We explain why our CG-GAN can improve the training stability from this perspective. We focus on the classifier of CG-GAN in the discriminator (i.e., the objective function Equation (7)). Note that the value $l_y(x)$ in $\sigma_{\text{sce}}(x, y)$ (Equation (3)) can be written as a dot-product $f(x)^\top w_y$, where $f(x)$ is the feature embedding vector and $w_y$ is the weight vector of the classifier associated with class $y$ in the last fully connected layer. Suppose we sample $n$ real data samples from the given training data, and $n$ generated data samples from the generator. For each $1 \le i \le n$, let $x_i^r$ and $x_i^g$ denote the $i$-th real and $i$-th generate samples, respectively. In practice, we use the empirical cross-entropy loss to represent $C_d^{cg}$ in Equation (7): First, we define

$$L_w(i) = -\log \Pr(y|x_i^r) + [m + \log \Pr(y|x_i^g)]_+, \quad (11)$$

where $\Pr(y|x_i^r) = \frac{\exp\left(f(x_i^r)^\top w_y\right)}{\sum_{k=1}^K \exp\left(f(x_i^r)^\top w_k\right)}$ and $\Pr(y|x_i^g) = \frac{\exp\left(f(x_i^g)^\top w_y\right)}{\sum_{k=1}^K \exp\left(f(x_i^g)^\top w_k\right)}$. Then we define the empirical loss $\hat{C}_d^{cg} = \frac{1}{n}\sum_{i=1}^n L_w(i)$ for training the discriminator.

LEMMA 3.2. *For $\forall k \in \{1, \ldots, K\}$, the gradient of $L_w(i)$ is:*

$$\frac{\partial L_w(i)}{\partial w_k} = \begin{cases} G_r(i, k), & \Pr(y|x_i^g) < \exp(-m); \\ G_r(i, k) - G_g(i, k), & \Pr(y|x_i^g) \ge \exp(-m), \end{cases}$$

*where $G_r(i, k) = -f(x_i^r)\left(\mathbf{1}_{y=k} - \Pr(y|x_i^r)\right)$ and $G_g(i, k) = -f(x_i^g)\left(\mathbf{1}_{y=k} - \Pr(y|x_i^g)\right)$. Here, $\mathbf{1}_{y=k}$ is the indicator function that equals to 1 if $y = k$.*

**Intuitive analysis from Lemma 3.2:** Lemma 3.2 reveals that when the confidence of the classifier for the generated data surpasses $\exp(-m)$, the gradient of $L_w(i)$ should be equal to $G_r(i, k) - G_g(i, k)$. Moreover, once the confidence exceeds $\exp(-m)$ (i.e., $m - \sigma_{\text{sce}}(x, y) > 0$), the objective function described in Equation (7) encourages the classifier to exhibit low confidence on the generated data. Since Proposition 3.1 reveals that the classifier encourages the generator to learn the real joint distribution, $f(x_i^g)$ is guided to be close to $f(x_i^r)$ and suppressing the confidence of the classifier on the generated data implicitly affects the confidence on the real data. Note that the gradient norm of the cross-entropy loss in AC-GAN is always equal to $||f(x_i^r)||\left(\mathbf{1}_{y=k} - \Pr(y|x_i^r)\right)$, and the norm $||f(x_i^r)||$ of AC-GAN is encouraged to increase to obtain high confidence. Consequently, the norm of the gradient $||G_r(i, k) - G_g(i, k)||$ in our CG-GAN should be less than the norm of gradient in AC-GAN, which can alleviate the gradient explosion problem and thus improve the training stability. We place the proof of Lemma 3.2 in supplement.

## 3.3 Potential Issues in CG-GAN and Improvements

**Potential over-confidence issue.** To implement our proposed CG-GAN as described in Section 3.1, it is crucial to consider the optimal confidence function of Equation (3) when the training objective for the CG-GAN classifier reaches its global optimum. Specifically, we denote the optimal softmax cross-entropy function of Equation (3) as $\sigma_{\text{sce}}^*(x, y)$ and the corresponding optimal confidence function as $\Pr^*(y|x)$. We propose the following corollary:

COROLLARY 3.3. *When the training objective for the classifier of CG-GAN achieves the global optimum, the optimal confidence function $\Pr^*(y|x)$ may be any value between $\exp(-m)$ and 1.*

We place the proof to our supplement. Corollary 3.3 reveals the optimal confidence function of CG-GAN may be close to 1, which correspondingly means that the feature norm of the sample will become large. As discussed in Section 2, large feature norm can lead to unstable training. As shown in Figure 2, our experimental findings observed that the CG-GAN model exhibited instability after long training. This observation highlights the necessity of introducing a regularization term to effectively address the issue of over-confidence in CG-GAN.

**Classifier criteria**. Before addressing the issue of overconfidence in CG-GAN, it is also important to consider the classification criteria of the auxiliary classifier. First, we clarify two necessary conditions for a qualified classifier. Suppose $P(x) = \{p_1(x), p_2(x), \cdots, p_K(x)\}$ is the prediction distribution by the classifier with the softmax activation function. For any input data $x$ with label $y$, the classifier should satisfy:

$$\begin{align} \textbf{(i)} \quad p_y(x) &\ge 1/K; & (12) \\ \textbf{(ii)} \quad p_y(x) &\ge p_k(x) \text{ for any } k \ne y, & (13) \end{align}$$

where $p_y(x) = \frac{\exp\left(l_y(x)\right)}{\sum_{k=1}^K \exp\left(l_k(x)\right)}$ and $p_k(x) = \frac{\exp\left(l_k(x)\right)}{\sum_{k=1}^K \exp\left(l_k(x)\right)}$. Based on Corollary 3.3, we have $p_y(x) = \Pr^*(y|x) \ge \exp(-m)$. We just simply let $m \le \log(K)$, and then we have $p_y(x) = \Pr^*(y|x) \ge \exp(-m) \ge \frac{1}{K}$. As a consequence, the classifier of CG-GAN satisfies the condition (12). However it cannot guarantee the condition (13). In particular, the negative softmax cross-entropy in Equation (7) minimizes $p_y(x) = \Pr(y|x) = \frac{\exp\left(l_y(x)\right)}{\sum_{k=1}^K \exp\left(l_k(x)\right)}$ (i.e, minimizes $l_y(x)$ while maximize $l_{k \ne y}(x)$) on the generated data, which may lead to $p_y(x) < p_{y \ne k}(x)$ (i.e., $l_y(x) < l_{k \ne y}(x)$).

**Our proposed solutions.** To better control the behavior of the classifier on the generated data with respect to the two conditions (12) and (13), we define the following prior label distribution:

$$\tilde{P} = \Big[ \frac{1 - \exp(-m)}{K-1}, \ldots, \underbrace{\exp(-m)}_{\text{The } y\text{-th item}}, \ldots, \frac{1 - \exp(-m)}{K-1} \Big]. \quad (14)$$

From the assumption $m \le \log(K)$, we have $\exp(-m) \ge \frac{1}{K} \ge \frac{1 - \exp(-m)}{K-1}$; then we can minimize the KL-divergence between the prior label distribution $\tilde{P}$ and the prediction distribution $P(x)$ on the generated data to encourage the classifier to satisfy the two conditions (12) and (13). **Moreover, the KL-divergence term can serve as a regularization mechanism to help mitigate the aforementioned over-confidence issue.** Hence, we can add the expected value of $\text{KL}(P(x)||\tilde{P})$ or $\text{KL}(\tilde{P}||P(x))$ to Equation (7). To better balance the losses of the discriminator and generator, we also add the expected value of $-\text{KL}(P(x)||\tilde{P})$ or $-\text{KL}(\tilde{P}||P(x))$ to Equation (8). We propose **rCG-GAN**: the CG-GAN with the

reverse KL divergence $\mathsf{KL}(P(x)||\tilde{P})$. Coupled with the GAN losses, the objective functions of rCG-GAN are defined as follows:

$$\min_D L_D + \lambda_1 \cdot C_d^{cg} + \lambda_2 \cdot \mathbb{E}_{x,y\sim Q_{XY}}[\mathsf{KL}(P(x)||\tilde{P})]; \quad (15)$$

$$\min_G L_G + \lambda_1 \cdot C_g^{cg} - \lambda_2 \cdot \mathbb{E}_{x,y\sim Q_{XY}}[\mathsf{KL}(P(x)||\tilde{P})], \quad (16)$$

where $\lambda_1, \lambda_2 > 0$ are two given coefficients. Similarly, we can also have **fCG-GAN**: the CG-GAN with the forward KL divergence, where we just simply replace the term $\mathsf{KL}(P(x)||\tilde{P})$ with $\mathsf{KL}(\tilde{P}||P(x))$ in Equation (15) and Equation (16). In Section 4, we mainly focus on rCG-GAN since fCG-GAN often achieves similar experimental results.

**REMARK** 1. *Note that the regularization term, specifically the KL term, is explicitly applied to the generated data. Nonetheless, it is crucial to acknowledge that suppressing confidence in the generated data can indirectly influence confidence in the real data. This occurs as the classifier encourages the generator to learn the real joint distribution, as indicated in Proposition 3.1. Such a feedback mechanism may lead to a decreased feature norm in both the real and generated data, consequently improving the stability of the CG-GAN. We provide the experimental verification of this phenomenon in the supplement.*

## 3.4 Comparison with Closely Related Works

**Differences between CG-GAN and EBGAN.** EBGAN [36] introduces an energy-based formulation for unconditional GANs, whereas our CG-GAN is specifically designed to address the issues commonly encountered in training instability and overconfidence in Classifier-based cGANs. Unlike EBGAN, CG-GAN incorporates an auxiliary classifier and employs label information and the softmax cross-entropy function. By analyzing the gradient of the classification loss in CG-GAN, we can explicate why our model improves training stability. Moreover, EBGAN has a fundamentally different network structure for the discriminator, which is based on an auto-encoder.

**Analysis on ReACGAN.** ReACGAN [13] employs the normalization of both feature embeddings and weight vectors to address the collapse issue. However, this method may have limited improvement on the condition generation performance compared to AC-GAN. Furthermore, the absence of adversarial training on the classifier of ReACGAN may make it challenging to ensure the generator to effectively learn the real data distribution, consequently diminishing the diversity of the generated samples.

**Analysis on ADC-GAN.** To better explain the possible problems of the auxiliary discriminative classifier GAN (ADC-GAN) [11] in the practical optimization, we discuss the mathematical expression of the objective functions of ADC-GAN below. It introduces the "discriminative classifier" which can play both the roles of the classifier and discriminator. In particular, for a training dataset with $K$ classes, the discriminative classifier $l$ maps each $x \in X$ to $\mathbb{R}^{2K}$, where $K$ dimensions serve for the $K$ labels of real data, and the other $K$ dimensions serve for the $K$ fake labels of generated data. To distinguish the real and fake labels, we add the superscripts "$r$" (for real data) and "$g$" (for generated data) to each ground-truth label $y \in Y$. So "$l_{y^r}(x)$" denotes the $y^r$-th element of $l(x)$ corresponding to the real ground-truth label $y^r$, and similarly, "$l_{y^g}(x)$" denotes the $y^g$-th element of $l(x)$ corresponding to the fake ground-truth

label $y^g$. Given a data $x \in X$, $\mathsf{Pr}(y^r|x)$ denotes the confidence that it is a real data and has the label $y^r$; and $\mathsf{Pr}(y^g|x)$ can be defined similarly. Also, similar with Equation (3) we have

$$\mathsf{Pr}(y^r|x) = \frac{\exp\left(l_{y^r}(x)\right)}{\sum_{k=1}^{2K} \exp\left(l_k(x)\right)}; \quad (17)$$

$$\mathsf{Pr}(y^g|x) = \frac{\exp\left(l_{y^g}(x)\right)}{\sum_{k=1}^{2K} \exp\left(l_k(x)\right)}. \quad (18)$$

The discriminative classifier tries to minimize the following Equation (19) and Equation (20):

$$C_d = -\mathbb{E}_{x,y\sim P_{XY}}[\log\mathsf{Pr}(y^r|x)] - \mathbb{E}_{x,y\sim Q_{XY}}[\log\mathsf{Pr}(y^g|x)]; \quad (19)$$

$$C_g = -\mathbb{E}_{x,y\sim Q_{XY}}[\log\mathsf{Pr}(y^r|x)] + \mathbb{E}_{x,y\sim Q_{XY}}[\log\mathsf{Pr}(y^g|x)]. \quad (20)$$

Then we can simplify Equation (20):

$$C_g = -\mathbb{E}_{x,y\sim Q_{XY}}[\log\mathsf{Pr}(y^r|x)] + \mathbb{E}_{x,y\sim Q_{XY}}[\log\mathsf{Pr}(y^g|x)]$$
$$= -\mathbb{E}_{x,y\sim Q_{XY}}[\log\mathsf{Pr}(y^r|x) - \log\mathsf{Pr}(y^g|x)]$$
$$= -\mathbb{E}_{x,y\sim Q_{XY}}[\log \frac{\exp\left(l_{y^r}(x)\right)}{\sum_{k=1}^{2K} \exp\left(l_k(x)\right)} - \log \frac{\exp\left(l_{y^g}(x)\right)}{\sum_{k=1}^{2K} \exp\left(l_k(x)\right)}]$$
$$= -\mathbb{E}_{x,y\sim Q_{XY}}[l_{y^r}(x) - l_{y^g}(x) -$$
$$\left(\log \sum_{k=1}^{2K} \exp\left(l_k(x)\right) - \log \sum_{k=1}^{2K} \exp\left(l_k(x)\right)\right)]$$
$$= -\mathbb{E}_{x,y\sim Q_{XY}}[l_{y^r}(x) - l_{y^g}(x)]. \quad (21)$$

In the optimization discriminator phase, the loss function, i.e., Equation (19), guides the classifier to act as a discriminator by distinguishing differences between real and generated data. However, the dependency between generated data and the real label is absent since the same class of real and generated data must be segregated into two distinct classes, namely the "real" and "fake" labels. This dependency is only provided during generator optimization via the loss function represented by Equation (21). Nevertheless, due to the absence of a cross-entropy loss function when optimizing the generator in Equation (21), the conditional generation ability of ADC-GAN may not be sufficiently trained.

## 4 EXPERIMENTS

We compare our proposed rCG-GAN (and fCG-GAN) with several recently proposed cGANs with BigGAN [1] backbone, including AC-GAN[30], PD-GAN [28], ReACGAN [13] and ADC-GAN [11]. We use the open-source library StudioGAN repository to conduct our experiments [1].

**Datasets and evaluation metrics.** We consider five public datasets: CIFAR10 [21] (60k images of 10 classes), CIFAR100 [21] (60k images of 100 classes), Tiny-ImageNet [23] (120k images of 200 classes), Baby/Papa/Grandpa-ImageNet [14] (each has 100 classes), and ImageNet [3] (1,281k and 50k images for training and validation with 1k classes). Baby/Papa/Grandpa-ImageNet are created by StudioGAN [14] for small-scale ImageNet experiments.

We consider six evaluation metrics: "Inception Score (IS)" [32], "Fréchet Inception Distance (FID)" [9] , "Density" and "Coverage" [29], and the "improved Precision" and "improved Recall" [22]. IS and FID are widely used metrics for evaluating the performance of

---

[1] https://github.com/POSTECH-CVLab/PyTorch-StudioGAN

generative models. Some studies [1, 37] show that IS tends to measure the generation fidelity and FID tends to capture the diversity of the generated data. Density and Coverage are a pair of metrics designed to disentangle the fidelity and diversity measurement from FID. The improved Precision and Recall [22] are proposed to overcome the shortcomings of the original Precision and Recall for the generated distribution against the real data distribution.

**Some experimental details.** We use the validation dataset as the default reference distribution for the computing of evaluation metrics. For CIFAR10 and CIFAR100, we use the test dataset due to the absence of the validation dataset. We adopt the default configurations of the ReACGAN paper [13] in StudioGAN and follow [5, 11] to use the hinge loss [24] for the implementation of the GAN losses $L_D$ and $L_G$. The number of training iterations is set to 100, 000 for CIFAR10/CIFAR100 and 200, 000 for the other five datasets; the batch size is set to 64 if not specified. The parameters $\lambda_1$ and $\lambda_2$ for our rCG-GAN and fCG-GAN are set to 1.0 (we also investigated the performance of rCG-GAN with different values of $\lambda_1$ and $\lambda_2$, and we place the results to our supplement due to the space limit).

## 4.1 Experimental Results

Note that the performances of rCG-GAN and fCG-GAN are similar in practice, so we mainly focus on rCG-GAN in our experiments except for the ablation study. Recall that our proposed CG-GAN depends on the parameter $m$; in fact the value $\exp(-m)$ is the desired confidence for $\Pr^*(y|x)$. So for convenience, the value $\exp(-m)$ is called by "desired confidence" in our experiments. According to our discussion in Section 3.3, we should set $\exp(-m) \geq 1/K$ for each testing dataset. Particularly, we also study the experiment with varying the value $\exp(-m)$ at the end of this section.

**Ablation study.** We conduct the experiments to show that the necessity of coupling the CG-GAN with the $\mathsf{KL}(P(x)||\tilde{P})$ or $\mathsf{KL}(\tilde{P}||P(x))$ (as discussed in Section 3.3), for improving the performance of CG-GAN and avoiding early-training collapse. As shown in Table 1, both fCG-GAN and rCG-GAN outperform CG-GAN in terms of six metrics on CIFAR10 and CIFAR100, and we observe that rCG-GAN performs slightly better than fCG-GAN with respect to FID. In the second part of our ablation study, we further examine the effectiveness of CG-GAN, fCG-GAN, and rCG-GAN in terms of training stability. Let "AC-GAN + rKL" and "AC-GAN + fKL" denote the methods of AC-GAN with $\mathsf{KL}(P(x)||\tilde{P})$ and $\mathsf{KL}(\tilde{P}||P(x))$, respectively. As shown in Figure 2, our CG-GAN, which is equipped with the adversarially trained auxiliary classifier, can achieve better stability comparing with the baselines "AC-GAN + rKL", "AC-GAN + fKL", ACGAN [30] and AMGAN [38]. Moreover, our proposed rCG-GAN and fCG-GAN can successfully avoid early-training collapse and exhibit superior stability compared to the basic CG-GAN.

**Comparison with existing cGANs.** We illustrate our experimental results on CIFAR-10, CIFAR-100, Tiny ImageNet and Baby /Papa/Grandpa-ImageNet in Table 2. From the results we can see that our rCG-GAN achieves the best of 5 scores (except for the improved Recall) on the datasets: the scores of IS, Density and improved Precision are used for measuring the generation fidelity; FID and Coverage are used for measuring the diversity of the generated images. So the results shown in Table 2 suggest that our proposed

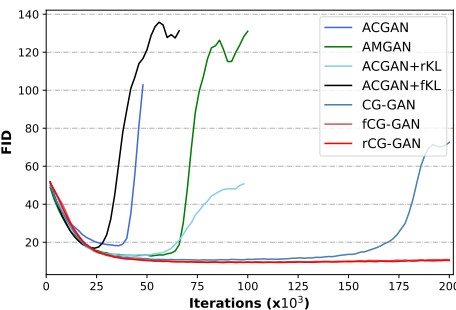

**Figure 2: FID on CIFAR-100.**

method yields better generation fidelity and diversity which are important metrics for evaluating the performance of generative model. Some qualitative results of our rCG-GAN and baseline approaches are shown in supplement due to the space limit.

**Image generation on ImageNet.** To further evaluate the performance of our method for large-scale dataset, we also consider the experiment on ImageNet, which is a widely used dataset in the computer vision community. We conduct the experiment on ImageNet with $128 \times 128$ image resolution in the batch size of 256, and the results are shown in Table 3. Our rCG-GAN outperforms other cGANs by a large margin in terms of IS and FID; moreover, we also report the classification accuracies and our advantage is also significant. The results indicate that rCG-GAN not only achieves high-quality image generation but also demonstrates promising performance in terms of conditional image generation. We also place several generated images in our supplement.

**Conditional generation performance on datasets with different levels of classification difficulty.** To have a more comprehensive comparison of the conditioning performance between rCG-GAN and other cGANs, we evaluate the Top-1 and Top-5 classification accuracies on generated images using the pretrained Inception-V3 network [34] on three subsets of ImageNet: Baby/Papa/Grandpa-ImageNet. These subsets were created by StudioGAN [14] based on the classification difficulty of the images. Baby-ImageNet represents the easiest subset to classify, while Grandpa-ImageNet represents the most difficult subset. From the results shown in Table 4, we can see that our rCG-GAN can achieve the best conditional generation performance in terms of classification accuracies as well as FID, across the datasets with different classification difficulty levels.

**Impact of the desired confidence.** In practice, we set $\exp(-m)$ to be slightly larger than $\frac{1}{K}$, where $K$ represents the number of classes in the dataset. By this setting, we can satisfy both condition (12) and (13) to guide the conditional generation in the training procedure. Additionally, a moderately small value of $\exp(-m)$ leads to a high entropy distribution in (14), that is helpful to mitigate the over-confidence issue. We conduct the experiments with varying the desired confidence $\exp(-m)$. As shown in Figure 3, our experiments on CIFAR100 indicate that the change of $\exp(-m)$ yields a mild trade-off between Improved Precision/IS and Improved Recall: the Improved Recall decreases as the Improved Precision and IS score increases, when we vary $\exp(-m)$ from 0.011 to 0.05. For the

**Table 1: The ablative results on CIFAR10 and CIFAR100. The best results are shown in bold. "↑" indicates higher is better, and "↓" indicates lower is better.**

| Datasets | Methods | IS↑ | FID↓ | Density↑ | Coverage↑ | Precision↑ | Recall↑ |
|---|---|---|---|---|---|---|---|
| CIFAR10 | CG-GAN | 9.999 | 8.194 | 1.044 | 0.9309 | 0.7697 | 0.6734 |
|  | fCG-GAN | 10.272 | 7.701 | 1.082 | 0.9356 | 0.773 | **0.675** |
|  | rCG-GAN | **10.285** | **7.514** | **1.109** | **0.9396** | **0.7759** | 0.6736 |
| CIFAR100 | CG-GAN | 13.42 | 11.459 | 1.009 | 0.8872 | 0.7904 | 0.5892 |
|  | fCG-GAN | **14.291** | 9.505 | 1.048 | 0.9233 | 0.8008 | **0.6307** |
|  | rCG-GAN | 14.0678 | **9.46** | **1.0721** | **0.9248** | **0.8092** | 0.6222 |

**Table 2: Evaluation on CIFAR10, CIFAR100, Tiny-ImageNet, Baby/Papa/Grandpa-ImageNet.**

| Datasets | Methods | IS↑ | FID↓ | Density↑ | Coverage↑ | Precision↑ | Recall↑ |
|---|---|---|---|---|---|---|---|
| CIFAR10 | PD-GAN | 9.969 | 8.004 | 1.068 | 0.9255 | 0.7587 | 0.6835 |
|  | AC-GAN | 9.936 | 8.342 | 1.031 | 0.9132 | 0.7549 | 0.6567 |
|  | ADC-GAN | 9.9903 | 8.0266 | 1.0003 | 0.9233 | 0.7496 | **0.6997** |
|  | ReACGAN | 9.841 | 8.026 | 1.056 | 0.9275 | 0.7711 | 0.6537 |
|  | rCG-GAN | **10.285** | **7.514** | **1.109** | **0.9396** | **0.7759** | 0.6736 |
| CIFAR100 | PD-GAN | 11.9238 | 10.8121 | 0.8599 | 0.866 | 0.7396 | 0.6992 |
|  | AC-GAN | 11.597 | 12.777 | 0.8945 | 0.8295 | 0.7519 | 0.5927 |
|  | ADC-GAN | 11.7254 | 10.7903 | 0.8566 | 0.8804 | 0.7358 | **0.7040** |
|  | ReACGAN | 12.1006 | 12.1964 | 0.9591 | 0.8372 | 0.7624 | 0.5783 |
|  | rCG-GAN | **14.0678** | **9.46** | **1.0721** | **0.9248** | **0.8092** | 0.6222 |
| Tiny-ImageNet | PD-GAN | 11.119 | 32.782 | 0.5519 | 0.5318 | 0.6258 | 0.6147 |
|  | AC-GAN | 11.092 | 36.799 | 0.5027 | 0.4591 | 0.6092 | 0.5141 |
|  | ADC-GAN | 12.932 | 26.682 | 0.5881 | 0.6012 | 0.6365 | **0.658** |
|  | ReACGAN | 13.0780 | 30.4484 | 0.6608 | 0.5589 | 0.6669 | 0.5051 |
|  | rCG-GAN | **19.657** | **16.83** | **0.8965** | **0.8146** | **0.7344** | 0.5981 |
| Baby-ImageNet | PD-GAN | 23.0264 | 32.0833 | 0.6179 | 0.6477 | 0.6553 | 0.7253 |
|  | AC-GAN | 27.071 | 27.453 | 0.7044 | 0.6611 | 0.6993 | 0.6603 |
|  | ADC-GAN | 24.2711 | 30.813 | 0.6069 | 0.6661 | 0.6515 | **0.7331** |
|  | ReACGAN | 27.2747 | 27.5857 | 0.7316 | 0.6487 | 0.7217 | 0.6213 |
|  | rCG-GAN | **31.5075** | **21.4124** | **0.7792** | **0.7644** | **0.7289** | 0.6831 |
| Papa-ImageNet | PD-GAN | 16.6445 | 34.6244 | 0.5827 | 0.6358 | 0.6212 | 0.6646 |
|  | AC-GAN | 22.15 | 30.701 | 0.7368 | 0.6808 | 0.7014 | 0.5226 |
|  | ADC-GAN | 18.7525 | 33.8927 | 0.5877 | 0.6606 | 0.6260 | **0.666** |
|  | ReACGAN | 20.2521 | 29.6279 | 0.7967 | 0.6708 | 0.7208 | 0.5350 |
|  | rCG-GAN | **26.9556** | **23.4174** | **0.8396** | **0.8086** | **0.7288** | 0.6352 |
| Grandpa-ImageNet | PD-GAN | 14.9834 | 30.0774 | 0.6714 | 0.7062 | 0.635 | 0.605 |
|  | AC-GAN | 18.639 | 28.899 | 0.7761 | 0.743 | 0.6782 | 0.5256 |
|  | ADC-GAN | 14.3486 | 31.2384 | 0.6472 | 0.6884 | 0.6296 | **0.6356** |
|  | ReACGAN | 18.0457 | 28.2561 | 0.8461 | 0.7458 | 0.6902 | 0.5012 |
|  | rCG-GAN | **22.445** | **22.679** | **0.9006** | **0.856** | **0.7248** | 0.579 |

experiments on other datasets, please refer to the details in our supplement.

## 5 RELATED WORK

PD-GAN [28] is a representative projection-based cGAN that incorporates class information into the discriminator by learning an embedding for each class. As a representative classifier-based cGAN, AC-GAN [30] uses an auxiliary classifier appended to the discriminator, and added the cross entropy loss to the standard GAN loss. Zhou et al. [38] introduced the AM-GAN, which employs a $(k + 1)$-way classifier along with extra "fake" labels for supervised learning. TAC-GAN [6] introduces a twin classifier to address the biased learning objective of AC-GAN. ContraGAN [12] applies the conditional contrastive loss and the cross-entropy loss to capture the data-to-data relationship and the data-to-label relationship. ECGAN [2] presents a comprehensive outlook on cGANs

**Table 3: Evaluation on ImageNet. Iters. means the training iterations. Top-1 Acc. and Top-5 Acc. mean the Top-1 and Top-5 classification accuracies (%) on the generated images using the pre-trained Inception-V3 network, respectively. \*: the results reported by the each original paper ; †: the results reported by [13]; ‡: the results reported by [14].**

| Methods | Iters. | IS↑ | FID↓ | Top-1 Acc. ↑ | Top-5 Acc. ↑ |
|---|---|---|---|---|---|
| TAC-GAN* [6] | - | 28.86 | 23.75 | - | - |
| StyleGAN2‡ [19] | - | 22.54 | 33.40 | 17.97 | 38.17 |
| StyleGAN3-t‡ [17] | - | 21.06 | 36.51 | - | - |
| PD-GAN† [28] | | 28.63 | 24.68 | 29.994 | 53.842 |
| AC-GAN† [30] | | 62.99 | 26.35 | 62.412 | 84.899 |
| ContraGAN† [12] | 200k | 25.25 | 25.16 | 2.866 | 11.482 |
| ReACGAN† [13] | | 50.30 | 16.32 | 23.210 | 51.602 |
| ADC-GAN [11] | | 38.972 | 20.415 | 37.347 | 60.495 |
| rCG-GAN | | **151.215** | **5.961** | **77.355** | **93.160** |
| PD-GAN† [28] | | 43.97 | 16.36 | - | - |
| ReACGAN† [13] | 500k | 68.27 | 13.98 | - | - |
| ADC-GAN* [11] | | 66.96 | 11.65 | - | - |
| rCG-GAN | | **173.319** | **5.187** | **79.872** | **93.656** |

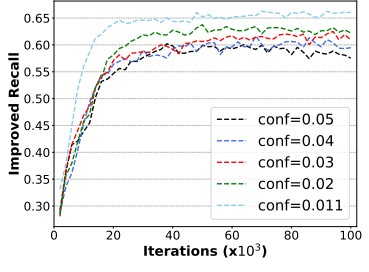

(a) Improved Recall curves.

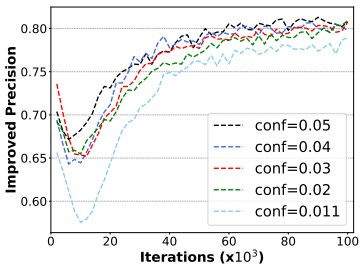

(b) Improved Precision curves.

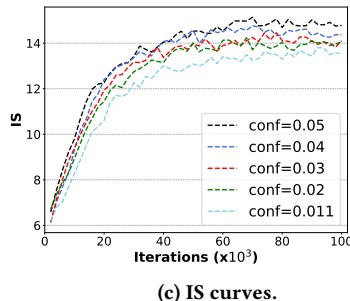

(c) IS curves.

**Figure 3: The desired confidence (conf) yields a mild trade-off between Improved Recall and IS/Improved Precision.**

**Table 4: Baby/Papa/Grandpa-ImageNet classification accuracies on generated images from cGANs.**

| | Methods | FID↓ | Top-1 Acc. ↑ | Top-5 Acc. ↑ |
|---|---|---|---|---|
| Baby | PD-GAN | 32.0833 | 45.551 | 64.047 |
| | AC-GAN | 27.453 | 56.208 | 75.444 |
| | ADC-GAN | 30.813 | 49.378 | 68.016 |
| | ReACGAN | 27.5857 | 51.409 | 70.306 |
| | rCG-GAN | **21.4124** | **62.527** | **80.223** |
| Papa | PD-GAN | 34.6244 | 22.44 | 42.08 |
| | AC-GAN | 30.701 | 33.98 | 59.00 |
| | ADC-GAN | 33.8927 | 26.02 | 46.36 |
| | ReACGAN | 29.6279 | 26.84 | 48.96 |
| | rCG-GAN | **23.4174** | **43.62** | **67.6** |
| Grandpa | PD-GAN | 30.0774 | 16.44 | 36.4 |
| | AC-GAN | 28.899 | 28.74 | 55.78 |
| | ADC-GAN | 31.2384 | 17.56 | 36.76 |
| | ReACGAN | 28.2561 | 19.74 | 44.58 |
| | rCG-GAN | **22.679** | **35.14** | **63.54** |

by considering both cGANs with and without classifiers. Zhou et al.

[37] introduced a novel approach that merges the discriminator with the classifier to create a multi-label classifier with $K + 2$ dimensions. MH-GAN [20] enhances AC-GAN by substituting the cross-entropy loss with a multi-class extension of the popular hinge loss. Kang et al. [13] proposed ReACGAN that normalizes both the feature embeddings and the weight vectors to avoid the collapse issue, and expanded the cross-entropy loss to the data-to-data cross-entropy loss. Hou et al. [11] introduced the method ADC-GAN that applies an auxiliary discriminative classifier to help the classifier for distinguishing real data from fake data.

## 6 CONCLUSION

In this paper, we propose a novel stable training method to improve the performance and stability of classifier-based cGANs, the key idea is to design an efficient adversarial training strategy for the auxiliary classifier and mitigate the over-confidence issue caused by the classifier. The experimental results suggest that our method not only provides improved training stability, but also produces high-quality generation and exhibits better conditional generation performance compared to several state-of-the-art cGANs on a set of popular benchmark datasets.

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
