# OpenReview forum: "A Novel Confidence Guided Training Method for Conditional GANs with Auxiliary Classifier"
_acmmm.org/ACMMM/2024/Conference — MM2024 Poster_

### Official Review · Reviewer_g5sU · 2024-05-20

**Rating:** 4
**Confidence:** 3

**Summary:**

This paper introduces a novel stable training method for conditional Generative Adversarial Networks (cGANs) to preserve generation fidelity and diversity, addressing the common issue of mode collapse. By incorporating adversarial training into a K-way classifier, this method aims to enhance the generator's ability to learn the real joint distribution. Extensive experiments demonstrate the effectiveness of the proposed method.

**Strengths:**

1. Introduced a novel confidence guided generative adversarial network (CG-GAN) and provides a solid theoretical analysis demonstrating how the proposed method encourages the generator to learn the real joint distribution of data.
2. Incorporated a prior label distribution and KL divergence term into the loss function, solving the problem of instability of CG-GAN after long-term training.

**Limitations:**

1. From the experimental results, this method seems to work well, and perhaps the open source code can be provided.
2. Why can adding KL divergence to the loss function solve the shortcomings of CG-GAN? Please provide a more detailed introduction or proof.
3. How does this method solve the problem of mode collapse? The author should propose corresponding experiments and theories to elaborate on this.
4. I would like to know the additional time and computational cost brought by this method.

**Suitability:**

2

---

### Official Review · Reviewer_xnx3 · 2024-05-22

**Rating:** 5
**Confidence:** 4

**Summary:**

The authors propose a novel and stable training method to improve the performance and stability of classifier-based conditional Generative Adversarial Networks (cGANs). The key idea is to design an efficient adversarial training strategy for the auxiliary classifier. They also propose to establish a high-entropy prior label distribution for the generated data and inposie a reverse KL divergence regularization term. The introduced method helps to mitigate the over-confidence issue caused by the classifier.

**Strengths:**

- The high-level idea of this paper is insightful. Introducing the adversarial training to the auxiliary classifier is novel and makes sense.
- The paper is well-written and easy to follow.
- The experiments are comprehensive, and the experimental results show that the proposed method can significantly outperform the relevant methods.

**Limitations:**

- My main concern is the proof for proposition 3.1. In Equ. (24), the optimal Q_{XY} is obtained via the classification loss. However, as shown in Equ. (7), it is also necessary to consider L_G to obtain the optimal Q_{XY}. As the authors claimed for their high-level idea, 'This intuition is somewhat similar to the adversarial strategy for training a standard GAN [7],' I am interested in seeing a proof like the one in [7].
- The authors claim that the proposed method can mitigate the over-confidence issue, but the paper lacks experimental results showing the classification confidence scores to demonstrate this.

**Suitability:**

3

---

### Official Review · Reviewer_EhLb · 2024-05-26

**Rating:** 3
**Confidence:** 3

**Summary:**

The paper addresses the issue of mode collapse and unstable performance in conditional Generative Adversarial Networks (cGANs) equipped with an auxiliary classifier. The proposed method introduces Confidence Guided Generative Adversarial Networks (CG-GAN), which leverage adversarial training strategies for the classifier to preserve generation fidelity and diversity. By integrating a high-entropy prior label distribution and a reverse KL divergence term into the minimax loss, the method enhances the performance and stability of cGANs. Extensive experiments on benchmark datasets like ImageNet demonstrate the advantages of this approach over existing state-of-the-art cGAN models.

**Strengths:**

**S1 Comprehensive Testing:** The effectiveness of CG-GAN is demonstrated through rigorous testing across multiple datasets, including large-scale benchmarks like ImageNet. The use of widely recognized metrics (Inception Score and Fréchet Inception Distance) further validates the improvements in both diversity and fidelity of generated images.

**S2 Theoretical Support:** The paper provides a theoretical basis for the proposed methods, explaining how the changes to the loss functions and the integration of adversarial training can lead to better learning of the real joint distribution of data and labels. This theoretical insight is crucial for understanding the impact of the proposed modifications.

**S3 Practical Improvements:** The proposed CG-GAN shows significant improvements over existing methods in terms of image quality and training stability. This is particularly important for applications requiring reliable and diverse image generation, such as in media creation or data augmentation for training other machine learning models.

**Limitations:**

**W1 Novelty Concerns:**

Auxiliary Classifier in GANs: The concept of integrating an auxiliary classifier within GAN architectures is not new. The Auxiliary Classifier GAN (AC-GAN) introduced by Odena et al. in 2016 ("Conditional image synthesis with auxiliary classifier GANs.") already utilized this concept to improve the quality of generated images by conditioning on class labels. The proposed CG-GAN method enhances this idea by introducing adversarial training for the classifier and a novel loss function involving reverse KL divergence, but the foundational concept remains similar to previous works. This might raise questions about the degree of innovation in the approach.
Improvements on Existing Concepts: While the paper introduces enhancements like high-entropy label distribution and specific loss adjustments for stability and fidelity, these are fundamentally incremental improvements over existing classifier-based GAN frameworks. For example, the use of adversarial training in the context of GANs has been extensively explored in literature, and while the specific application to the classifier component is somewhat novel, it builds directly on established methods.

**Suitability:**

2

---

### Official Review · Reviewer_jYoz · 2024-05-27

**Rating:** 3
**Confidence:** 4

**Summary:**

This paper introduces a novel conditional generative adversarial network training method called Confidence Guided Generative Adversarial Networks (CG-GAN), aiming to address the issue of mode collapse in existing cGANs training processes and enhance the diversity and quality of generated samples. The authors devise an efficient adversarial training strategy tailored to the auxiliary classifier and propose a novel high-entropy prior label distribution, enhancing the stability and performance of CG-GAN through a reverse KL divergence term. Experimental results on multiple public datasets demonstrate significant advantages of CG-GAN over state-of-the-art cGANs in terms of generation quality and diversity evaluation metrics such as Inception Score (IS) and Fréchet Inception Distance (FID), particularly showcasing remarkable performance on large-scale dataset ImageNet. Additionally, the paper explores the impact of expected confidence on model performance and demonstrates its robustness across datasets with varying levels of classification difficulty through experiments. Overall, this work provides valuable insights and methodologies for improving the training stability and generation performance of cGANs.

**Strengths:**

1. Innovation: The paper introduces a novel cGAN training method, namely CG-GAN, which addresses mode collapse and overconfidence issues by introducing adversarial training with an auxiliary classifier. This represents a novel research direction in the field of GANs.
2. Theoretical Analysis: In addition to proposing a new method, the author provides in-depth theoretical analysis, including demonstrating that the auxiliary classifier in CG-GAN encourages the generator to learn the conditional real joint distribution, thereby enhancing the academic depth of the study.
3. Experimental Validation: The paper conducts experimental validation on multiple widely recognized benchmark datasets, including the large-scale ImageNet dataset. The experimental results indicate that the proposed method exhibits advantages in both generation quality and diversity.

**Limitations:**

1. In Figure 1(a), the trend of rCG-GAN continues to rise within 200 iterations. Will it keep increasing with further iterations? Why doesn't the graph extend to the convergence point of rCG-GAN? Does it keep increasing indefinitely?
2. Technical Transparency: The paper may not sufficiently elaborate on the technical details of the new method, particularly regarding the specific implementation of adversarial training strategies and the design choices for the classifier. Such information is crucial for replicating experiments and further research.
3. Unclear Structure: The paper's structure lacks clarity, and the main theme is not sufficiently defined. Related work and the innovative contributions of the paper are intertwined, making it difficult to emphasize the key points.
4. Lack of Visual Representation: As a paper proposing a novel model, there is a lack of visual aids such as model diagrams or flowcharts for overall clarification.
5. Generalization Verification: The experiments mainly focus on specific cGAN architectures, lacking sufficient discussion on the generalization capabilities of the new method across different types of cGAN architectures or various application scenarios.
6. Computational Efficiency: Although the new method shows excellent performance in improving generation quality, the paper lacks discussion on computational efficiency and resource consumption. This is crucial for evaluating the practicality and scalability of the method.
7. The paper should provide generated instances within the main text. Existing evaluation metrics for generation do not adequately assess the real visual effects. Generated instances aid in better evaluating the performance of the method.

**Suitability:**

2

---

### Meta-Review · Area_Chair_xSzb · 2024-06-28

**Recommendation:** Accept (Poster)
**Confidence:** 4

**Metareview:**

After the rebuttal, three reviewers (EhLb, xnx3, and g5sU) reached a consensus to accept the paper due to its innovation, extensive experimental results, and in-depth theoretical analysis. Reviewer jYoz assigned a Borderline Reject because of doubts about the paper's innovation, but still agrees to the paper being accepted. The rebuttal effectively addressed the major concerns raised by the reviewers. After reading the paper, its rebuttal, and the discussion, the AC finds the paper well-written, sound, and well-justified by the experiments. Thus, the AC recommends accepting the paper.